# RFID and Drones: The Next Generation of Plant Inventory

Jannette Quino [1], Joe Mari Maja [2,*], James Robbins [3], R. Thomas Fernandez [4], James S. Owen Jr. [5] and Matthew Chappell [6]

1. Edisto Research and Education Center, 64 Research Road, Blackville, SC 29817, USA; jquino@clemson.edu
2. Department of Agricultural Science, Clemson University, 240 McAdams Hall, Clemson, SC 29634, USA
3. University of Arkansas Div. of Ag. CES, 2301 S. Univ. Ave., Little Rock, AR 72204, USA; jrobbins@uaex.edu
4. Department of Horticulture, Michigan State University, 1066 Bogue St., East Lansing, MI 48824, USA; fernan15@msu.edu
5. USDA-ARS Application Technology Research Unit, 1680 Madison Ave., Wooster, OH 44691, USA; jim.owen@usda.gov
6. Horticulture Department, University of Georgia, 326 Hoke Smith Building, Athens, GA 30602, USA; hortprod@uga.edu
* Correspondence: jmaja@clemson.edu

**Abstract:** Collection of plant inventory (i.e., count, grade, plant size, yield) data is time-consuming, costly, and can be inaccurate. In response to increasing labor costs and shortages, there is an increased need for the adoption of more automated technologies by the nursery industry. Growers, small and large, are beginning to adopt technologies (e.g., plant spacing robots) that automate or augment certain operations, but greater strides must be taken to integrate next-generation technologies into these challenging unstructured agricultural environments. The main objective of this work is to demonstrate merging specific ground and aerial-based technologies (Radio Frequency Identification (RFID), and small Unmanned Aircraft System (sUAS)) into a holistic systems approach to address the specific need of moving toward automated on-demand plant inventory. This preliminary work focuses on evaluating different RFID tags with respect to their distance and orientation to the RFID reader. Fourteen different RFID tags, five distances (1.5 m, 3.0 m, 4.5 m, 6.0 m, and 7.6 m), and four tag orientations (the front of the tag (UP), back of the tag (DN), tag at sideways left (SL), and tag at sideways right (SR)) were assessed. Results showed that the tag upward orientation resulted in the highest scanning total for both the laboratory and field experiments. Two orientations (UP and SR) had significant effect on the scan total of tags. The distance between the reader and the tags at 1.5 m and 6.0 m did not significantly affect the scanning efficiency of the RFID system in horizontally fixed ($p$-value > 0.05) position regardless of tags. Different tag designs also produced different scan totals. Overall, since most of the tags were scanned at least once (except for Tag 6F), it is a very promising technology for use in nursery inventory data acquisition. This work will create a unique inventory system for agriculture where locations of plants or animals will not present a barrier as the system can easily be mounted on a drone. Although these experiments are focused on inventory in plant nurseries, results for this work has potential for inventory management in other agricultural sectors.

**Keywords:** RFID; drone; microcontroller; ornamental; precision agriculture; inventory

## 1. Introduction

Ornamental crop production is one of the fastest-growing segments of U.S. agriculture. In 2017, the U.S. nursery industry sales were USD 5.9 billion, a 15.4% increase from 2012 (USDA NASS 2019). The global ornamental market was valued at USD 48 billion in the last two years and is expected to increase by more than 54% by the end of 2026. Nursery production is labor intensive and vulnerable to rising labor expenses and possible future labor shortages. In 2016, according to data from Agricultural Resource Management Survey, labor costs accounted for 14% of U.S. agriculture's total operating expenses, but as much as 39% of total expenses for nursery production. Surveys continue to document a shortage

of labor for the nursery industry. Forty-three surveyed growers (field and container) in 2011 unanimously indicated the need for improved inventory data (plant count and grade). One-third of participating growers stated they collect inventory data at least twice per year at an estimated cost of 2.8% of gross sales and yet current methods still result in sales losses due to poorly timed or inaccurate inventory data. The survey estimated that the nursery industry spends over USD 31 million annually on labor to collect inventory data. This entails high labor cost, excessive hours, exorbitant overhead expenses, and inaccurate data due to human error [1].

Small unmanned aircraft systems (sUAS) are quickly evolving into a useful platform for a variety of agricultural tasks including detecting diseases and weeds [2,3], yield prediction [4], water stress [5] and spraying chemicals [6]. Images collected by sUAS has been used to validate models using statistical analysis [5] and most recently using artificial intelligence [7].

A Radio Frequency Identification (RFID) tag functions as a barcode that can hold information [1]. There are two different categories of RFID tags: active, and passive, which can be made in different shapes and forms [1,8] for varying applications. Passive tags are made of paper, plastic, or vinyl [8,9]. RFID tags are available that can tolerate continuous exposure to outdoor nursery production environments including water, heat, dirt and chemicals. RFID labels provide a unique identification code for every label and can be encoded with important production information such as genus, species, cultivar, planting dates, etc. Passive RFID labels are inexpensive, and as their adoption increases, costs will decrease as a matter of scale. Passive RFID labels are read with a particular UHF radio frequency using a RFID interrogator. An advantage in agricultural operations is that RFID does not require a clean label unlike commonly used barcodes [10]. RFID tags have been used in farms for tagging and tracking animals, plants and health monitoring [11], identify and track livestock [10] and monitoring of the irrigation system management [12]. RFID systems have been used in the food industry and medical field [13]. Ma et al. [14] investigated using sUAS combined with Radio Frequency Identifiers (RFID) for inventory control in warehouses. Bunker and Elshebeni [15] developed a small portable RFID scanner comprised of a Raspberry Pi, and interrogator. They used a chip similar to what was used, but theirs included the development kit where their code will only interface directly to the USB port of the development kit. Their system was designed to be highly customizable and modular. Bridge et al. [16] developed similar system using an Arduino-based RFID platform where its application was for three animal applications (breeding behaviours of Wood Ducks, RFID-enabled bird feeder, and nest-box monitoring for breeding birds). Their system used an Arduino M0 and two RFID module, UB22270 from Atmel. The RFID modules used a simplified antenna, thin coil magnet wire, which can be easily created with an inductance between 1.25~1.3 mH. The antenna was placed on the entrance hole such that the movement of tagged ducks will be recorded. Although, the system was quite unique as it used a small RFID chip, its applications are very similar to RFID applications in manufacturing, where the interrogator is placed in one location. Recent work evaluating RFID and drones for monitoring and management of animals is reported [17], although the technology was used for different purposes. RFID was only used for position and tracking while the drone was used for counting animals using cameras. A study reported using RFID and wireless technology to predict the moisture content of rice [18]. Both the Received Signal Strength Indicator (RSSI) from the two wireless transceivers were used for predicting the moisture content in rice using Artificial Neural Network. They reported that both the RSSI of RFID and Zigbee transceivers can be used to predict the moisture content with an accuracy of more than 85%.

The overriding objective of this research effort is to develop and promote sustainable strategies that improve the profitability of the nursery industry, e.g., to demonstrate merging specific ground- and aerial-based technologies (sUAS and RFID) in a whole system approach to address the specific need of providing near on-demand plant inventory. The focus of this effort is:

1.   To evaluate a RFID tracking system to scan and identify RFID tags using a sUAS;
2.   To identify different RFID tags and their efficiency with respect to the distance and orientation relative to the RFID reader.

## 2. Materials and Methods

### 2.1. Study Site and the RFID System

Research was conducted at the Edisto Research and Education Center (EREC), located in Blackville, SC, USA (33.3570N, 81.3271W). The overall studies were designed to evaluate a RFID tracking system to scan and identify RFID tags using a sUAS. The RFID system consisted of the RFID tag, reader module, the dashboard application, and sUAS (Matrice 600 Pro, Shenzhen DJI Sciences and Technologies Ltd., Shenzhen, China) (Figure 1). The maximum payload capacity for this sUAS is 6 kg using six TB47S batteries which is twice the weight of the RFID reader system (2.3 kg). Most of the weight of the RFID reader system was due to the antenna, which can be minimized in future experiments by changing to a lighter antenna.

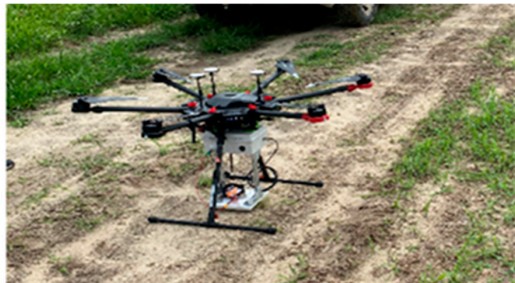

**Figure 1.** Radio Frequency Identification (RFID) reader and DJI Matrice 600 Pro.

### 2.2. The RFID Tag and Tag Bytes

RFID tags used in these experiments were passive RFID tags. Passive tag designs can vary in shape and size. To identify each tag design used in this experiment, tag design codes were assigned (e.g., TDx where TD stands for Tag Design and x the different tag inlays). Table 1 includes a list of tag designs evaluated in this experiment. The RFID Tag or Electronics Product Code (EPC) byte is a 12-bit unique tag identifier code encoded on each RFID tags. It can be obtained through a RFID reader or the RFID dashboard application. Some RFID tag manufacturers provided multiple copies of the same tag design, but each tag has a unique EPC.

**Table 1.** RFID passive tag designs representing variation in inlay patterns and sizes.

| Tag Design Code | Tag Image | Tag Design Code | Tag Image |
|---|---|---|---|
| TDA |  | TDE |  |
| TDB |  | TDF |  |
| TDC |  | TDG |  |
| TDD |  | TDH |  |

### 2.3. The RFID Reader Module

The RFID Reader Module (RFID-RM) used in this study is composed of the ARM Cortex-M4 based core controller (MK66FX1M0VMD18, NXP, Eindhoven, The Netherlands), reader chip (m6e, Jadak, NY, USA) with a 7 dB patch antenna, wireless transceivers (Holybro Telemetry Radio V3, Holybro, Wanchai, Hong Kong, China), microSD card socket, and GPS module (MTK3339, GlobalTop Technology Inc., Taiwan). The ARM Cortex-M4 comes with a 256 Kb Static Random-Access Memory (SRAM), 1280 Kb of Flash RAM, 4 Kb of EEPROM, 6 UART, 3 SPI, 4 i2c, 2 USB controllers and 1 Ethernet port. It also has 100 programmable GPIO pins with 25 16-bit Timer and 4 32-bit Timer. The RFID-RM retained the high-performance ARM Cortex and can be custom programmed to read the reader chip and employ different communication protocols to transfer data to and from the transceivers and GPS module. The customized main board has two voltage regulators; one for the controller and the other for the reader chip as indicated in Figure 2. It has a provision to be connected to a radio control (R/C) receiver that can be used to trigger the power of the reader chip through a R/C transmitter (see Figure 2 (R/C receiver port)). A ground pad is located on the bottom of the reader chip to minimize noise and function as a heat sink. The reader chip, m6e, is configured to use the highest power and work at a frequency of 900 Mhz. The communication between the m6e and controller is through a serial port with 115,200 baud rate. It is configured to use GEN2 Tag protocol and the region was set to North America. All these and other optional configurations (power, number of antennas, microSD card, baud rate for debug, and antenna port used) are updated at every power reboot as the m6e does not have non-volatile memory that can save the prior configurations. At boot up, information is streamed to the transceivers and can be viewed through the dashboard as well.

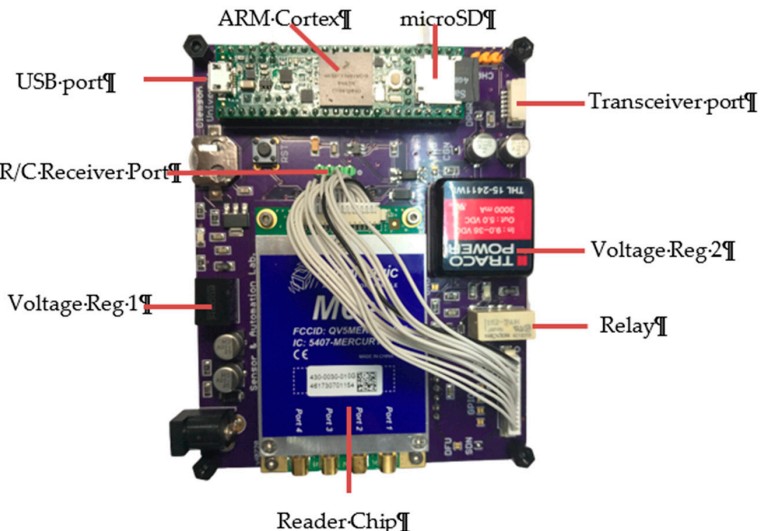

**Figure 2.** The RFID Reader Module (RFID-RM) with key components labeled.

The RFID-RM was designed to include a real-time clock unit and is powered through a coin-cell battery (1.5 V) located below the USB port. Time was used as part of the file naming process for data whenever the system was powered. All data are saved to a unique file and stored in the microSD card. The same data are also streamed through the transceiver port. Data are typically stored in a comma separated values (CSV) file. The RFID-RM used polling of tags at regular interval at 10~15 ms sleep mode was not implemented in the current firmware as it would only be powered in a short span of time. If a tag is detected, it will extend the duration of the polling time to read the tag ID. The frequency and duration of the reading can be changed through the firmware.

An off-the-shelf plastic electronic enclosure (dimensions) was modified to house the electronics as shown in Figure 3a. The power switch, microSD socket, transceiver antenna,

RFID antenna, and GPS connector were routed outside the box for easy access (see Figure 3b). The RFID antenna was attached to the box by aluminum extrusion bars. Space (25 cm) between the bottom of the electronics box and the antenna was chosen to provide clearance for connectors and yet not interfere with landing the sUAS.

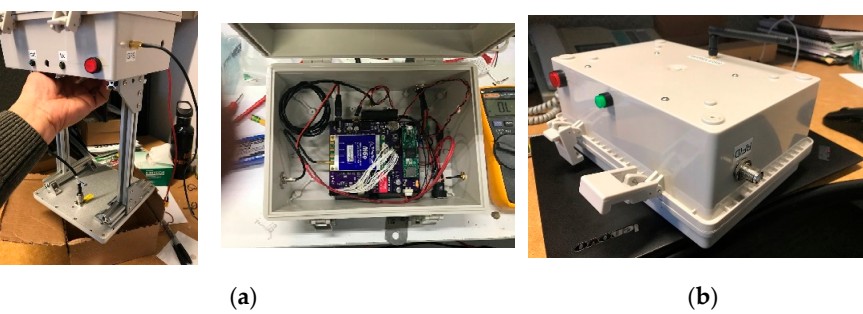

(**a**)                                                                           (**b**)

**Figure 3.** (**a**) RFID enclosures, antenna, and (**b**) connector for GPS and transceivers.

The RFID-RM can be powered by a single Lithium Polymer battery with at least 1000 mAh from 9 V~16 V. The battery used was a 12 V with 2200 mAh (3 Cell).

### 2.4. The RFID Dashboard

The RFID dashboard displays RFID tag information on the computer screen. The dashboard was developed using Java (JRE, Oracle Corporation, Austin, TX, USA). The RFID system application has two dialogs, the port configuration setting, and the main dashboard display. The port configuration setting will check for any existing open ports. It has a dropdown menu for the user to select port numbers and the correct baud rate. The main dashboard displays the tag information in real time. Tag information such as GPS latitude and longitude, received signal strength indicator (RSSI), phase, electronic product code (EPC), number of bytes received, time the tag was read, EPC tag bytes, input voltage, RFID voltage, and temperature (Figure 4). Other information displayed on the dashboard are users' configuration settings (port number and baud rate), RFID reader settings, RFID reader time, and the summary of all the data captured. The text box on the right can display initial settings of the RFID reader and the tag information. It has a user option to save the document in a specific folder, by clicking the "Click to save file" button. The initial settings of the RFID reader can be acquired by running the dashboard application before starting the RFID reader.

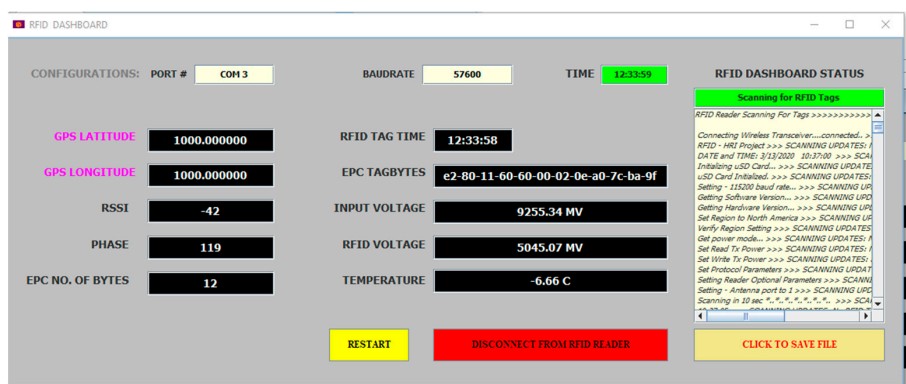

**Figure 4.** The RFID dashboard display. On the right of the display is the text file showing the RFID reader settings.

### 2.5. Laboratory Experiment

2.5.1. Experiment 1: Static Assessment of Total Tag Scan

The first experiment was a static assessment of 14 types of RFID tags. For three tag designs (TDA, TDB, TDC) listed in Table 1. Duplicate copies (each with a unique code) of

each tag were included in this experiment. Thus, for TDA, three copies (coded 5B, 5A, 59) were included. For design TDB, two copies (39, 80) and for design TDC, two copies (BF, 9F) were included. The RFID reader and 14 tags were placed on a wood easel as shown in Figure 5. In the first experiment, two factors were evaluated: distance between the reader and tag and orientation of the tag relative to the reader (Table 2). Five different distances were evaluated (treatments): 1.5 m, 3.0 m, 4.5 m, 6.0 m, and 7.6 m. Four different tag orientations were evaluated for each distance: upward (UP; i.e., tag upper surface facing directly toward the receiver), downward (DN; i.e., tag flipped over so the upper face of the RFID tag is now pointing away from the receiver), Sideways Left (SL), and Sideways Right (SR). The RFID reader was placed in a stationary position on its side and elevated 0.61 m above the floor, with the antenna pointing directly towards the tags (Figure 6). For each treatment (Table 3), the reader was turned on for 6.5 min to collect tag readings. The firmware written in the RFID-RM was configured to read all RFID tags energized nearby. This instruction allows each tag to rescanned as many times as possible during the predetermined testing period. In real life settings once the tag is scanned it declines further access.

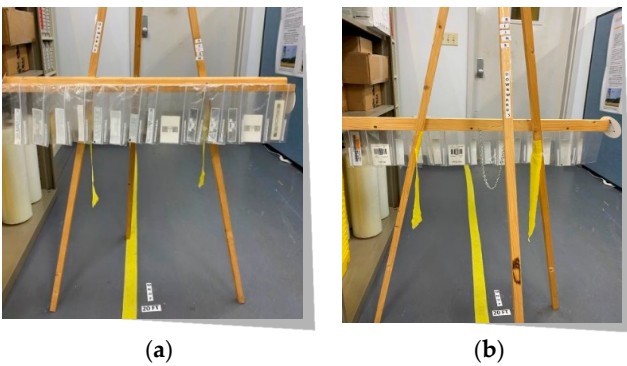

(**a**)                                                     (**b**)

**Figure 5.** Tags positioned in an (**a**) upward orientation and (**b**) downward orientation (i.e., upper surface pointed away from reader).

**Table 2.** Factors (2) evaluated in the first laboratory experiment for a total of nine treatments.

| Distances | Tag Orientations |
|-----------|------------------|
| 1.5 m | Front of the tag directed to the RFID-RM antenna (UP) |
| 3.0 m | Back of the tag directed to the RFID-RM antenna (DN) |
| 4.5 m | Tag at sideways left directed to the RFID-RM antenna (SL) |
| 6.0 m | Tag at sideways right directed to the RFID-RM antenna (SR) |
| 7.6 m | |

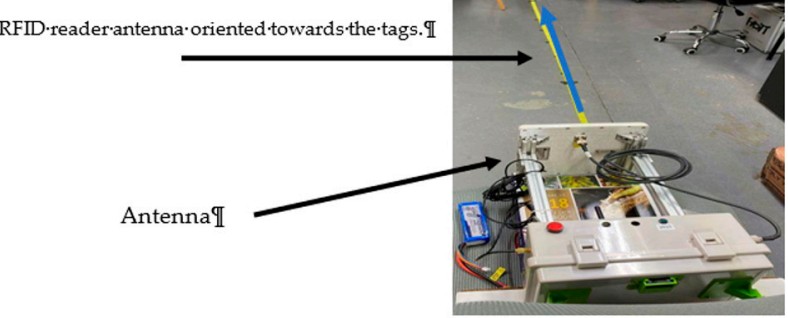

**Figure 6.** RFID-RM scanning tags during the first laboratory experiment.

**Table 3.** The treatment combinations for laboratory experiment 2.6.2.

| Treatment | DISTANCE | TAG Orientation | Treatment | DISTANCE | Tag Orientation |
|-----------|----------|-----------------|-----------|----------|-----------------|
| A | 1.5 m | Facing upwards (UP) | K | 4.5 m | Sideways Left (SL) |
| B | 1.5 m | Facing downwards (DN) | L | 4.5 m | Sideways Right (SR) |
| C | 1.5 m | Sideways Left (SL) | M | 6 m | Facing upwards (UP) |
| D | 1.5 m | Sideways Right (SR) | N | 6 m | Facing downwards (DN) |
| E | 3 m | Facing Upwards (UP) | O | 6 m | Sideways Left (SL) |
| F | 3 m | Facing downwards (DN) | P | 6 m | Sideways Right (SR) |
| G | 3 m | Sideways Left (SL) | Q | 7.6 m | Facing upwards (UP) |
| H | 3 m | Sideways Right (SR) | R | 7.6 m | Facing downwards (DN) |
| I | 4.5 m | Facing Upwards (UP) | S | 7.6 m | Sideways Left (SL) |
| J | 4.5 m | Facing downwards (DN) | T | 7.6 m | Sideways Right (SR) |

### 2.5.2. Experiment 2

Experiment 1 was designed to screen 14 tags to determine those with the highest scan total under specific laboratory conditions. Based on that experiment, four tags [9F (TDC), 6F (TDE), 5B (TDA), and 12 (TDI)] with the highest scan total were used in this experiment. This experiment evaluated 2 factors (tag type x tag orientation) with three replications. Each treatment was scanned for 3 min. Figure 7 shows a sample treatment orientation for treatment M, N, O and P at 6.0 m away from the RFID reader. Table 4 is the Randomized Complete Block Design treatment. To measure the variation in data, the standard error equation (Equation (1)) was used.

$$S.E = \sqrt{\frac{\sum_{s=1}^{m}\sum_{i=0}^{n}\gamma is2}{(n\gamma - 1)(ny)}} \tag{1}$$

where:

$s$ = series number
$i$ = point number in series s
$m$ = number of series for point y
$n$ = number of points in each series
$\gamma is$ = data value of series s and the ith point
$ny$ = total number of data values in all series

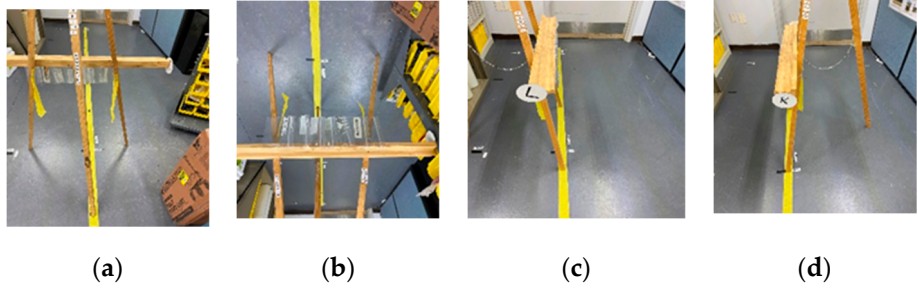

| (a) | (b) | (c) | (d) |

**Figure 7.** Examples of tag orientations at 6.0 m for M, N, O, and P. (**a**) M at upwards (**b**) N at downwards (**c**) O at Sideways Left (**d**) P at Sideways Right.

**Table 4.** The treatment design for laboratory experiment.

| REPLICATIONS | | | | | TREATMENTS | | | | | | | | | | | | | | | |
|---|---|---|---|---|---|---|---|---|---|---|---|---|---|---|---|---|---|---|---|---|
| 1 | H | M | Q | G | B | N | I | R | A | L | D | C | T | S | F | O | E | J | P | K |
| 2 | L | E | J | F | D | G | Q | H | C | O | M | P | S | T | K | N | B | I | A | R |
| 3 | Q | P | H | T | F | B | I | K | D | M | E | R | C | G | N | A | L | O | J | S |

*2.6. Field Experiment*

For field experiments, the RFID-RM system was attached to the underside of the sUAS. There were two sub-experiments in the field: (1) scanning tags while moving horizontally at a fixed altitude across tags on the ground (Vertically Fixed or VF), and (2) holding the same horizontal position over tags and then scanning tags at three altitudes (Horizontally Fixed or HF).

2.6.1. Vertically Fixed (VF)

The first field experiment was designed to evaluate the scan total of 6 tag designs when the aircraft would fly back and forth at a fixed height over tags on the ground (VF). Two factors were evaluated: altitude and tag orientation (Table 5). The experiment was designed so data for all 7 treatments could be collected during the same flight (single battery). The three altitudes evaluated were: 4.5 m, 6.0 m, and 7.6 m. Moreover, four different tag orientations were used for each altitude: tag facing upward (UP), downward (DN), Sideways Left (SL), and Sideways Right (SR). To ensure that a complete experimental run could be completed using a single aircraft battery, duplicate copies of some tag designs (5B = 5A = 59; BF = 9F; 2F = 80 = 39) were used in this experiment. Tags used were: 5A, BF, 5B, 2F, 59, 12, 81 and 6F. Tag 81 was included since it yielded a high scan total in laboratory experiment when orientated sideways to the reader. These eight RFID tags were taped to a 5 cm × 10 cm wood board in four different orientations. The tags designated in the upward (UP) orientation were: 5A and BF, while for the downward (DN) orientation were: 5B and 2F. Furthermore, the tags designated for Sideways Left (SL) were: 59 and 12. Tag 81 and 6F was designated at Sideways Right (SR). Wood boards were placed 3 m apart from each other in a parallel configuration.

**Table 5.** Factors evaluated in the field experiment.

| Altitude | Tag Orientation |
|----------|-----------------|
| 4.5 m | Facing Upwards |
| 6 m | Facing downwards |
| 7.6 m | Sideways Left |
|  | Sideways Rights |

Figure 8 shows the experimental layout. Tags which were attached to a 5 cm × 10 cm board were placed in the middle of road. Four meters on either side of the center of the road and in line with the boards were two markers. Markers were used to help the pilot stay centered while flying over the tags and served as visual markers for the outer boundary of aircraft flight. As an example, the aircraft would start a run outside the southern marker, fly in a straight line over the tags and continue beyond the northern most marker. One flight path is equivalent to one complete trip from south to north and vice versa. This process was repeated 5 times for each of the 3 altitudes at a forward flight speed of approximately 6 km per hour. Tags were equally scanned through the number of flight paths.

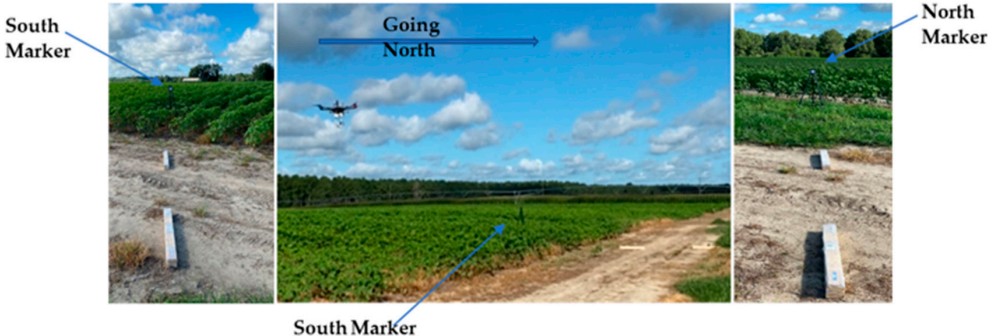

**Figure 8.** Illustration of aircraft flight path with tags in the center.

### 2.6.2. Horizontally Fixed (HF)

In this experiment, the sUAS was flown with a fixed horizontal position (Figure 9) but at three different altitudes (4.5 m, 6.0 m, and 7.6 m). Each altitude was scanned for one minute.

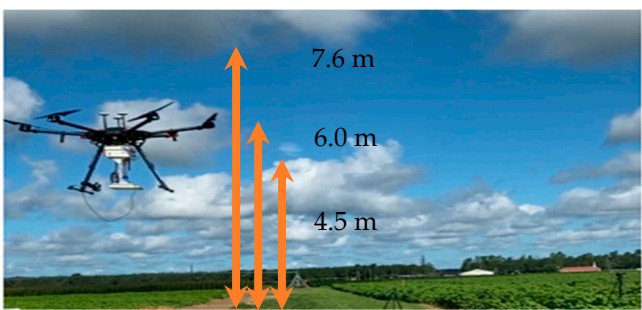

**Figure 9.** RFID-RM reader Small unmanned aircraft systems (sUAS) positioned directly over tags on the ground and then the aircraft was flown directly above those tags at three altitudes.

### 3. Results and Observations

#### 3.1. Static Assessment of Total Tag Scan

The total number of scans for each tag at the four distances (1.5 m, 3 m, 4.5 m, 6 m, and 7.6 m) over 6.5 min is shown in Figure 10. When the five distances and four orientations were combined, four tags (9F, 6F, 5B, and 12) had the highest number of scanned.

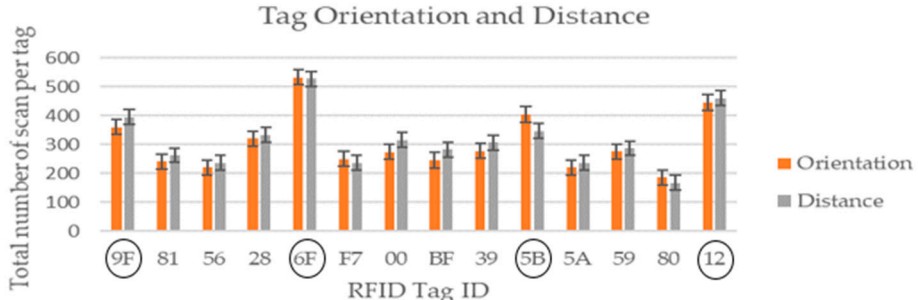

**Figure 10.** Results for the total number of scans per tag over 6.5 min.

#### 3.2. Laboratory Experiment 2

Tags 9F, 6F, 5B, and 12, respectively, were used for this experiment. Results in Figure 11 shows the summary of the number of tags scanned by orientation. The SR had the highest number of tags scanned, while the UP had the least number of tags scanned. Figure 12 summarizes the number of tags scanned by distance. The distance with the highest number of tags scanned is at 1.5 m, while 7.6 m had the least number of tags scanned. The result of the orientation table shows the standard deviation and the *p*-value of the tag 9F, 6F, 5B, and 12 on the data collected (Table 6). Each tag orientation was scanned at a distance from 1.5 m to 7.6 m. The signal response between Tags 9F and 6F was not significantly different (*p*-value > 0.05) for all 4 orientations. Moreover, tags 5B and 12 at UP and SR orientation produced a significant result (*p*-value < 0.05).

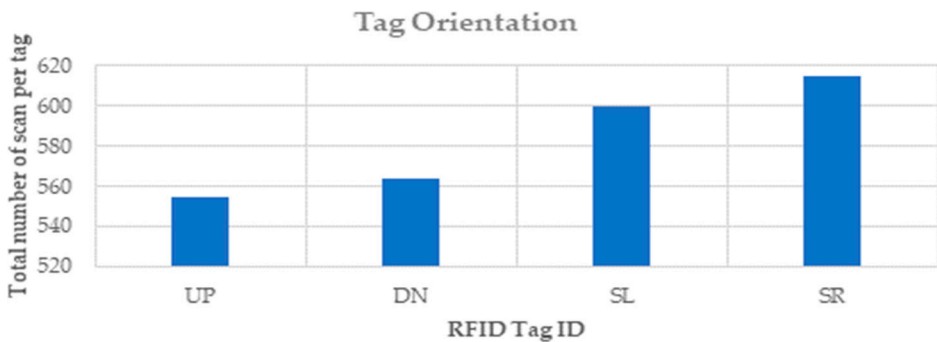

**Figure 11.** The total number of scans per tag for each type and orientation per 3 min in a laboratory.

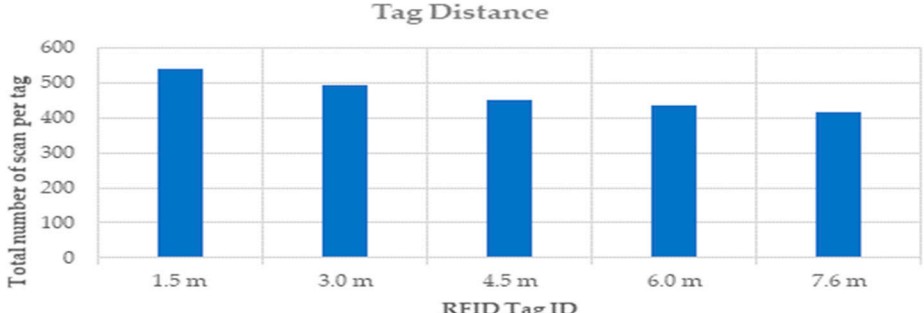

**Figure 12.** The total number of scans per tag for each type and distance per 3 min in a laboratory.

**Table 6.** Experimental results of the Orientation.

| Tag | Orientation | MEAN | SD | *p*-Value |
|---|---|---|---|---|
| | Upwards | 7 | 3.91 | 0.8646 |
| 9F (TDC) | Downwards | 7 | 5.33 | 0.1516 |
| | Horizontal 90° sideways L | 9 | 3.05 | 0.0611 |
| | Horizontal 90° sideways R | 11 | 3.35 | 0.5818 |
| | Upwards | 15 | 4.56 | 0.7136 |
| 6F (TDE) | Downwards | 12 | 5.66 | 0.1400 |
| | Horizontal 90° sideways L | 12 | 3.46 | 0.8851 |
| | Horizontal 90° sideways R | 13 | 1.82 | 0.3036 |
| | Upwards | 5 | 3.26 | 0.0032 * |
| 5B (TDA) | Downwards | 5 | 2.63 | 0.0799 |
| | Horizontal 90° sideways L | 7 | 4.63 | 0.9140 |
| | Horizontal 90° sideways R | 8 | 4.11 | 0.0490 * |
| | Upwards | 10 | 4.61 | 0.0007 * |
| 12 (TDI) | Downwards | 14 | 4.63 | 0.4443 |
| | Horizontal 90° sideways L | 12 | 3.90 | 1.000 |
| | Horizontal 90° sideways R | 9 | 3.00 | 0.0484 * |

* Indicates significant result.

The result of the distance table shows the standard deviation and the *p*-value of the tags 9F, 6F, 5B, and 12 (Table 7). At each distance, tags were scanned in all 4 orientations. The following tags generated a significant result (*p*-value < 0.05): 9F at 3.0 m, both 6F and 5B at 4.5 m, and tag 12 at 7.6 m.

**Table 7.** Experimental results of the distance.

| Tags | Distance | MEAN | SD | *p*-Value |
|---|---|---|---|---|
| | 1.5 m | 11 | 3.57 | 0.7962 |
| | 3.0 m | 8 | 3.82 | 0.0125 * |
| 9F (TDC) | 4.5 m | 8 | 4.46 | 0.0789 |
| | 6.0 m | 9 | 3.12 | 0.6570 |
| | 7.6 m | 8 | 5.37 | 0.2305 |
| | 1.5 m | 14 | 4.09 | 0.8660 |
| | 3.0 m | 13 | 5.02 | 0.3365 |
| 6F (TDE) | 4.5 m | 12 | 3.09 | 0.0487 * |
| | 6.0 m | 11 | 3.68 | 0.5496 |
| | 7.6 m | 13 | 4.57 | 0.3979 |
| | 1.5 m | 8 | 2.62 | 0.0610 |
| | 3.0 m | 8 | 3.52 | 0.6699 |
| 5B (TDA) | 4.5 m | 7 | 5.05 | 0.0153 * |
| | 6.0 m | 6 | 3.04 | 0.3535 |
| | 7.6 m | 2 | 2.17 | 0.1463 |
| | 1.5 m | 11 | 4.65 | 0.3633 |
| | 3.0 m | 12 | 4.09 | 0.0083 |
| 12 (TDI) | 4.5 m | 11 | 2.59 | 0.0668 |
| | 6.0 m | 11 | 5.46 | 0.5670 |
| | 7.6 m | 11 | 5.12 | 0.0190 * |

* Indicates significant result.

### 3.3. Field Experiment—Vertically Fixed (VF)

Figure 13 summarizes results for the number of tags scanned while the sUAS was moving horizontally at a fixed altitude across tags (VF). Tag 2F showed the highest number of total scans at 6 m followed by tag 59, then tags 5A and 5B. In addition, tag 6F at sideways left had the lowest number of scans. Figure 13 summarizes the effect of tag type and orientation on the number of total signals received over 1 min when the sUAS was flown at a fixed height above tags on the ground. 5A-U means 5A in an upward position and 2F-D means 2F in a downward position.

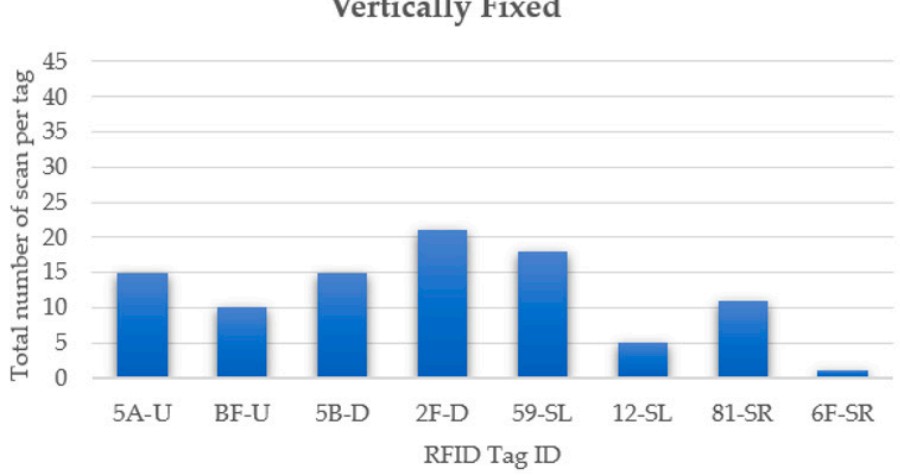

**Figure 13.** The total number of scans per tag type and orientation when the sUAS was flown back and forth at a fixed height over tags on the ground (VF) for 1 min.

### 3.4. Field Experiment—Horizontally Fixed (HF)

Figure 14 summarizes results for the total number of scanned received by a tag in a specified period of time (1 min.) while the sUAS was held at three altitudes above tags (HF). Tag 81 resulted in the largest number of scans followed by Tag 5A. In addition, Tag 6F has 0 readings.

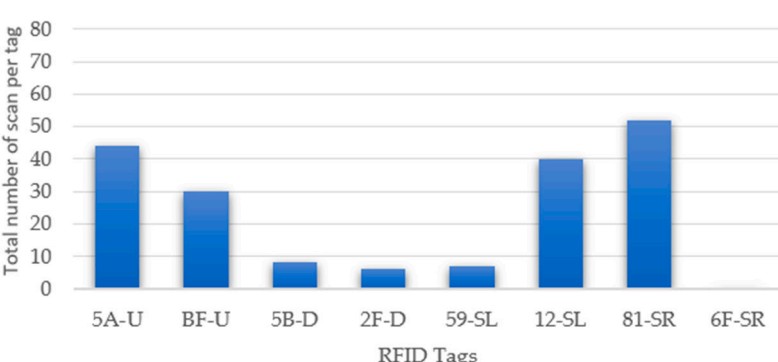

**Figure 14.** The total number of scans per tag type and orientation when the sUAS was flown with a fixed horizontal position at three different altitudes (4.5 m, 6.0 m, and 7.6 m) for 1 min.

## 4. Discussion

Tag scanning results varied depending on tag inlay type and size (Figures 13 and 14). Some of the tags scan rates were 70% lower than other tags. For example, Tag 80 total readings were 168, while 6F total readings were 527. Furthermore, Tag 6F had the highest number of scans and got the highest reading at SR and SL, while Tag 12 got the highest readings at DN and UP orientation. The highest number of scans at 7.6 m were tag 6F; at 6.0 m was Tag 9F; at 1.5 m was Tag 00, and at 3 m and 4.5 m was Tag 12. In addition, some tags inlay has a scan rate of 95% more, for example, at 7.6 m, Tag TDX (80) has 5 scans, while TDE (6F) has 231 scans.

In the second laboratory experiment, all four orientations of Tags 9F and 6F did not have a significant effect on the total number of scans ($p$-value > 0.05) which showed that the two-tag design have good performance. Fifty percent of the tags at UP and SR orientation shows a significant result ($p$-value < 0.05) for Tags 5B and 12. Moreover, the DN and SL produced insignificant results ($p$-value > 0.05).

Up to this point, the discussion has focused on comparing differences in the cumulative total of scans by different tags under both laboratory and field conditions. When the total is expressed as a rate (number of scans/unit time) an unexpected result is observed. If we use as an example Tag 6F, scan rate in the first experiment was 81 scans/min (527 total scans/6.5 min), while in the field experiment (VF), the same tag yielded 1 scan/min. A plausible explanation for this difference in the scan rate was due to the interference of signals (Remote Control, Transceivers, GPS) generated by various components of the sUAS system which were not an issue in the laboratory test.

While in the HF experiment, the RFID reader was able to scan all tags in each altitude and in all 4 orientations except for Tag 6F. The results on the data collected also showed a huge difference in comparison to the laboratory experiment with regard to the orientation. For example, the scan total of Tag 6F at horizontal 90° SR position was 0, but in the laboratory tag 6F has the highest number of scanned in the same orientation. Furthermore, the results on the data collected show a difference with regard to the distance as well. Tag 6F has 0 scan in all 3 altitudes, but in the laboratory tag 6F has the highest scan.

Two papers [15,16] report on an RFID system similar to ours. Bunker and Elsherbeni [15] built an integrated RFID system using the same RFID module we used. Our studies used the M6e chip while they used the development kit board of M6e. Our objective was to minimize the size of our RFID box and focus in developing our own drivers for the M6e. Although, their system was portable enough, it was not intended to be used where the RFID reader will be moving during operation as compared to our application. Bridge et al. [16] developed their own Arduino-based system and used two RFID chips from Atmel. Their system was unique as it used an easily fabricated antenna. While their antennae were small, the power of their system was very limited as their application was confined to small

space. Our effort summarizes critical foundational efforts that will be used in future field experiments at a large commercial nursery. We acknowledge a possible limitation of our current system in flight time; however, we envision improvements in battery technology will advance in the future which will address this limitation.

## 5. Conclusions

Based on our experiments the two-tag orientations that affect the scan total are the UP and SR, where SR also reveals to have the highest scan total. There is no significant effect on all tags at DN and SL orientation. In addition, tag design TDE at UP orientation holds the highest number of scanned while tag TDI in SR with the least number of scanned. Furthermore, the distance between the reader and the tags at 1.5 m and 6.0 m does not significantly affect the scanning efficiency of the RFID system in horizontally fixed ($p$-value > 0.05) in all tags. This preliminary work concludes that the RFID tag sensor patterns and inlay designs perform differently in a diverse state, some of the tag design scan total efficiency was very low while other tags were 95% more efficient compared to the other tag designs. The following factors that influence the RFID-RM scanning capabilities are the orientation of the tags, and distance. However, other factors such as the tag design, antenna angular sensitivity, etc., may also have an effect to the scan total. Overall, since most of the tags were scanned at least once (except for Tag 6F), it is a very promising technology for using in nursery inventory data acquisition. Future work for this project is to test a holistic system in commercial nurseries.

**Author Contributions:** Conceptualization, J.M.M., J.Q.; methodology, J.M.M., J.Q.; software, J.Q.; validation, J.Q., J.M.M.; formal analysis, J.Q., J.M.M.; investigation, J.Q., J.M.M.; resources, J.M.M., J.R., J.S.O.J., R.T.F., and M.C.; data curation, J.Q.; writing—original draft preparation, J.Q., J.M.M. and J.R.; writing—review and editing, J.M.M., J.R., J.S.O.J., R.T.F., and M.C.; visualization, J.Q.; supervision, J.M.M.; project administration, R.T.F.; funding acquisition, R.T.F., J.M.M., J.R., J.S.O.J. and M.C. All authors have read and agreed to the published version of the manuscript.

**Funding:** This research was partially supported by Horticultural Research Institute (HRI) Grant number 5935985 and is based on work supported by NIFA/USDA under project numbers MICL02473, and SC-1700543.

**Institutional Review Board Statement:** Not Applicable.

**Informed Consent Statement:** Not Applicable.

**Data Availability Statement:** Not Applicable.

**Acknowledgments:** The authors would like to thank Avery Dennison Corporation and R.A. Dudley Nurseries Inc. for their support and assistance in this project.

**Conflicts of Interest:** The authors declare no conflict of interest.

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
