# Peer review of "RFID and Drones: The Next Generation of Plant Inventory"

_agriengineering, doi:10.3390/agriengineering3020011_

Round 1

Reviewer 1 Report

Most of plants growers are very busy in their wide range of work, to achieve high standards of their final results. Presented tag system can be utilized during monitoring of the overall production concerning plants growing, plants harvesting, plants transportation and plants storage and distribution. The inventory data always should be very accurate during growing season, as well when selling field production to the final customer. Using flying objects as a reading units of scanning process, it can help to provide overall data collection system. This system can be utilized in the practice in nursery farms and other agricultural production sectors, where we can use tags for product identification porpouses.

Author Response

The authors are thankful for the reviewer’s comments and suggestions. We addressed all the comments and suggestions below with reference on our revised manuscripts when possible (line numbers):

  1. Most of plants growers are very busy in their wide range of work, to achieve high standards of their final results. Presented tag system can be utilized during monitoring of the overall production concerning plants growing, plants harvesting, plants transportation and plants storage and distribution. The inventory data always should be very accurate during growing season, as well when selling field production to the final customer. Using flying objects as a reading units of scanning process, it can help to provide overall data collection system. This system can be utilized in the practice in nursery farms and other agricultural production sectors, where we can use tags for product identification porpouses.

Response: The authors completely agree with the reviewer on these comments. Although, our results are preliminary, our next work is focused on testing our system at a large commercial nursery in the U.S.

Reviewer 2 Report

In Agriculture, the combined use of unmanned aerial vehicles and radiofrequency identification devices opens up a set of new application fields addressed to high speed data acquisition system in both greenhouse and open field. But, few and very preliminary investigations may be found over scientific literature. The objective of this this preliminary research focuses on evaluating different RFID tags with respect to their distance and orientation to the RFID reader.

In my opinion, the Authors have dealt extensively in descriptive aspects of the system they set up (12 figures and 6 tables), but they have missed their targets due to a questionable statistical analysis. For this reason, the results are unclear and often not conform to data. Herein, just few examples:

e.g Line 336 340 - The SR had the highest number of tags scanned, while downward orientation resulted in the lowest number of tags scanned. In addition, orientation at SR has the highest error, while the SL had the least error. Figure 14 shows the summary of the number of tags scanned by distance. The distance with the highest number of tags scanned is at 7.6 m, while 1.5 m has the least number of tags scanned.

From the figures, I can see just in four cases out of the fourteen SR (for Tag Orientation) and 7.5 (Tag Distance) returns the highest number.

e.g. L 350 and following. In Mat & Met the Authors report that in view of the results of Experiment 1, four tags [9F (TDC), 6F (TDE), 5B (TDA) and 12 (TDI)] with the highest scan rate were chosen. Unfortunately, in the results of experiment 1 they did not report any information about this. Moreover, I believe that the statistic method applied (it would have been more correct to say "not applied") does not allow the Authors a correct choice.

I added in the pdf text, by means of the adobe text tools, some more suggestions.

Author Response

The authors are thankful for the reviewer’s comments and suggestions. We addressed all the comments and suggestions below with reference on our revised manuscripts when possible (line numbers):

  1. In my opinion, the Authors have dealt extensively in descriptive aspects of the system they set up (12 figures and 6 tables), but they have missed their targets due to a questionable statistical analysis. For this reason, the results are unclear and often not conform to data. Herein, just few examples:

e.g Line 336 340 - The SR had the highest number of tags scanned, while downward orientation resulted in the lowest number of tags scanned. In addition, orientation at SR has the highest error, while the SL had the least error. Figure 14 shows the summary of the number of tags scanned by distance. The distance with the highest number of tags scanned is at 7.6 m, while 1.5 m has the least number of tags scanned.

From the figures, I can see just in four cases out of the fourteen SR (for Tag Orientation) and 7.5 (Tag Distance) returns the highest number.

e.g. L 350 and following. In Mat & Met the Authors report that in view of the results of Experiment 1, four tags [9F (TDC), 6F (TDE), 5B (TDA) and 12 (TDI)] with the highest scan rate were chosen. Unfortunately, in the results of experiment 1 they did not report any information about this. Moreover, I believe that the statistic method applied (it would have been more correct to say "not applied") does not allow the Authors a correct choice.

I added in the pdf text, by means of the adobe text tools, some more suggestions.

Response: Thank you for the comments and suggestions. We understand that the figure in the results were not clear about the top four tags mentioned in the Materials and Methods. We updated the figures to show the top four tags on the preliminary laboratory test. We also deleted the statistical analysis part as suggested by other reviewers and will address each of the comments in the attached pdf.

  1. Join Table 1 and

Response: We presumed that the reviewer wanted to join Table 1 and 3. But one of the reviewers suggested that we delete the Table 3 as it is not very important. Table 3 was deleted. We decided to keep Table 1 as is. We hope this address this suggestion.

  1. Join the figure 13 and 14

Response:  As suggested we joined Figure 3 and 4 in the revised version of the manuscript (line 188).

  1. Delete or join with figure 3-4

Response: This comment was referring to Figure 6, we deleted Figure 6 as suggested.

  1. Join tables 1 and 3. Delete EPC Tagbyte

Response: We deleted Table 3 as suggested by the other reviewers.

  1. Why did you report the experimental design in the paragraph’s title?

Response: This is to differentiate the first laboratory test. We revised the title to “Experiment 2”.

  1. Delete (in reference to Table 5)

Response: We kept Table 5 as per other reviewers’ suggestions but modified based on their input.

  1. Delete (in reference to Figure 10)

Response: We deleted Figure 10 as suggested.

  1. A robust statistic should (or even must) be chosen. Maybe a two-way Anova?

Response: We decided to delete this subsection as recommended by other reviewers.

  1. Somewhat is unclear. Just in four cases out of the fortheen SR (for Tag Orientation) and 7.5 (Tag Distance) returns the highest number.

Response: We apologize for not being clear with the statement. We  have now highlighted the plot for which of the four Tags have high total scanned in this experiment, but we only showed the complete plot. We updated the figure to clearly show the tags with the highest total scanned and also updated the narrative discussing this result as shown in Line 325-327.

  1. Highlighted – Randomized Complete Block Design

Response: We deleted the Randomized Complete Block Design to be consistent with the update in the Materials and Methods.

  1. In Mat & Met the Authors report that in view of the results of Experiment 1, four tags [9F (TDC), 6F (TDE), 5B (TDA) and 12 (TDI)] with the highest scan rate were choosen. Unfortunately, in the results of experiment 1 they did not report any information about this. Moreover, I belive that the statistic method applied (it would have been more correct to say "not applied") does not allow the Authors a correct choice.

Response: For this experiment, we have narrowed the number of tags and tag designs that we used for the succeeding experiment. We are interested in the effect of orientation and distance based on the four tags for this experiment. The narrative has been corrected to focus on which of the four orientation and five distances provide the highest scan based on the four tags used.

  1. Is this the acronym of Randomized Complete Block Design?

Response: Yes, it is, and we updated the title of the table.

  1. ??? It’s exactly the opposite, how the table's note highlights

Response: We apologize for this lapse, we agree with the reviewer, it should be the opposite, and this was corrected within the narrative (Line 360).

  1. Once again, I can not see in fig 17 what the Authors report

Response: We apologize for the misunderstanding. We changed the figure (Fig. 17 is now Fig. 13) to show the total number of scans for the three distances. We also deleted the orientation in the narrative to reduce redundancies.

  1. Is it positive? negative? What do you means for not have a significant effect? For example, I might consider that the reading from Tag 9F and 6F could not affected by orientation, and thus it is a good performance.

Response: We updated the sentence for the two tags (9F and 6F) to reflect that their orientation did not have a significant effect on scan. We also added the suggested statement by the reviewer that these tags gave good performance (Line 395~397).

  1. The Authors should try to explain the reasons of these differences

Response: We updated the narrative on the differences (Line 400~407).

Up to this point the discussion has focused on comparing differences in the cumula-tive total of scans by different tags under both laboratory and field conditions. When the total is expressed as a rate (number of scans/unit time) an unexpected result is observed. If we use as an example Tag 6F, scan rate in the first experiment was 81 scans/min (527 total scans/6.5 min), while in the field experiment (VF), the same tag yielded 1 scan/min. A plausible explanation for this difference in the scan rate was due to the interference of signals (Remote Control, Transceivers, GPS) generated by various components of the sUAS system which were not an issue in the laboratory test.

Reviewer 3 Report

See attached comments

Author Response

The authors are thankful for the reviewer’s comments and suggestions. We addressed all the comments and suggestions below with reference on our revised manuscripts when possible (line numbers):

  1. This work is in general original and interesting. Authors should better stress the relevance of the developed method, clarifying how in practice the UAV-RFID coupled systems can be used in agricultural applications.

Response: The authors already mentioned the relevance of this work to the different issues in ornamental industry, e.g., increasing labor cost and labor shortages, inaccurate and costly inventory. We also highlighted the statement in the Abstract on our holistic approach of using these two technologies (Line 22-25):

The main objective of this work is to demonstrate merging specific ground and aerial-based technologies (Radio Frequency Identification [RFID], and small Unmanned Aircraft System [sUAS]) into a holistic systems approach to address the specific need of moving toward automated on-demand plant inventory.

And finally, we added this in Line 35 ~ 37, to emphasize the relevance of our system to agriculture:

      This work will create a unique inventory system for agriculture where locations of plants or animals will not present a barrier as the system can easily be mounted on a drone.

  1. I the introduction the authors should explain how in practice the developed method can be applied to assist agricultural operations. Indeed simple identification can be done as the tags are positioned in the field. Thus the advantage of using UAVs is that the tag can be recognized everytime the UAVs fly in the field. So: why/when is this needed? For instance for plant specific treatments? Please report some applicative examples.

Response: Our focus for this work is mainly on ornamental plant inventory. Nurseries normally have a large ever-changing inventory of plants during the growing season (mostly outdoor) and most of these pots are moved to several locations during the production cycle. Due to the volume, it is very difficult to keep track where each of the plants are located at certain times. Real time inventory is critical in this industry, but most nurseries only perform inventory twice a year due to labor cost, shortage of labor (Line 57) and the high labor cost (Line 56). This work is focus on the preliminary work and our system will be field tested in a large commercial nursery in the U.S. Results of this work will provide the industry a timely, accurate and inexpensive inventory system with minimal labor as most of the data collected are automated. Data collected are both stored in the microSD card and transmitted via a transceiver.  

  1. The paper has many paragraphs/sub-paragraphs which are too short (2-4 lines): this makes the text fragmented and makes reading difficult. Please merge into larger paragraphs

Response: We revised the manuscript with the following:

  1. Both the Study site and RFID system subsection were combined (Line 109) and all subsections numbering were updated as highlighted in the manuscript.
  2. Both the subsections RFID Tag and Tag Bytes were combined (Line 132 ~ 140).
  3. Subsection RFID-RM Carrier was combined to Subsection 2.1 (Line 115~118)
  4. Please correct the title: 2.1. Subsection

Response: We updated Section 2.1 to Study site and the RFID System (Line 109)

  1. The monitored parameter is not clear. I understand that you were flying (horizontally and vertically) in the field and reading tags, but then I do not understand how you quantify the performance (what you call “number of tags”). Please clarify the point.

Response: We apologize for the confusion. In both field experiments, horizontal and vertical scanning, our objectives were the same as in the first two laboratory experiments: the effect of distance and orientation on the scan of each tag. Although, for inventory purposes in the real world, a single read for each tag is enough, our program in the RFID module keeps reading tag over the designated time period as long as the tag signal is received. We updated the label to “Total number of scan per tag” to avoid confusion. Note that we also affixed the orientation to the tag ID in this figure, for example, U-5A (U – means Upward) for Tag 5A was read 8 times at 6 meters (yellow) and updated the narrative accordingly (Line 370~373)

       Figure 13 summarizes the effect of tag type and orientation on the number of total signals received over 1 minute when the sUAS was flown at a fixed height above tags on the ground. 5A-U means 5A at Upward position and 2F-D means 2F at Downward position.

  1. The system has a weight of 3kg: this just half of the payload of the implemented UAV, but is in general high for standard UAVs. Please discuss in the paper if there are systems which allow reduction of such weight.

Response: The weight of our system (2.3 kgs) was due to the antenna. All our electronics, enclosure and even the LIPO battery that power the system was around 1 kg. We added the following sentence in the paper (Line 116~ 118):

      Most of the weight of the RFID reader system was due to the antenna which can be minimized in future experiments by changing to a lighter antenna.

  1. Energy consumption and duration of batteries is an issue for UAVs. In this case batteries has to be used also for the RFID antenna. Please discuss autonomy: how the RFID is affecting batteries duration, and in particular when larger distances are considered between the antenna and the tag.

Response: The RFID system developed in this work was powered separately from the UAV as indicated in Line 191 ~ 192:

The RFID-RM can be powered by a single Lithium Polymer battery with at least 1000mAh from 9V~16V. The battery used was a 12V with 2200mAh (3 Cell). 

We added as part of our Discussion the limitations of our system due to the battery (Line 425 ~ 427)

We acknowledge a possible limitation of our current system in flight time, however, we envision improvements in battery technology will advance in the future which will ad-dress this limitation.

  1. Table 3: EPC Tagbyte is a useless information. I would keep just the first two columns (maybe redrawing them horizontally). Alternatively the table can be eliminated and the information reported in plain text.

Response: We deleted Table 3 as suggested.

  1. Please organize Table 5 horizontally rather than vertically.

Response: We updated Table 5 as suggested.

  1. Forward flight speed, and presence of humidity (/rain water /frost) on the tag or canopy leaves over the tag may alter in a relevant way the reading performance. Have you tested this? Please comment.

Response: We know that water affects the performance of the tag and thus in our field experiments we made sure that there was no water on the tags as we attached it to the wood block. Our future experiments at a commercial nursery will focus on investigating the effect of environmental (moisture, dirt, foliage) on signal transmission. The purpose of these experiments was to narrow the number of variables investigated. In terms of flight speed, we tried to use a constant speed during this test. We will soon test the system in a commercial nursery setting where we will focus on the impact of aircraft speed and environmental parameters on signal transmission.

  1. In Figures 3 and 9: please avoid curved borders: just use normal borders: they are more scientifically sound. Please remove grey borders from graphs.

Response: We apologize for the borders. This was due to importing plot to the manuscript. We have updated the figures and removed the grey borders from graphs.

  1. In the graphs, I do not understand the meaning of “scanned per tags”.

Response: We apologize for the confusion. The vertical axis “No. of scanned per tag” is the number of times the reader receives the tag signal during the test. We updated the label to “Total number of scan per tag” to avoid confusion. We also addressed the same issue in No. 5.

  1. Figure 15 is corrupted

Response: We don’t know why this figure was corrupted on your end as other reviewers did not report the same issue, but we made sure on this revision that it is not corrupted.

  1. In case the paper will be accepted for revision, please address above comments and correct accordingly the paper, - giving your pertinent comments in the “Response to reviewer” document - reporting in the “Response to reviewer” document also the paragraph with amended text highlighted with yellow colour or the new amended figure.

Response: We have addressed each of your comments and highlighted the changes on our manuscript as suggested. All reviewers’ comments and suggestions were also highlighted in the same manuscript. Again, we thank you for the comments and suggestions which strengthens the manuscript.

Reviewer 4 Report

The title of the manuscript (MS) deals with "Radio Frequency Identification and small Unmanned Aircraft System: The next generation of plant inventory". The topic of this manuscript is of interest and well written and I liked reading it, great job!

Just a few comments.

  • Please consider spelling out the abbreviations (RFID and sUAS) in the article's title.
  • "In the "Introduction" section, this section can be improved to provide further background and include all relevant references. There is a need that you will use "recent publications" on the topic to make your research attractive. 
  • In the "discussion" section, the author must extend the comparison between their approach and other ones that have been developed and used in the literature for the same or related purposes (I recommend increasing the number of Scientific articles cited, especially to compare the study context with similar studies). Also, in this section, the authors should also highlight the current limitations and usefulness of the proposed research, and briefly mention some precise directions that they intend to follow in their future research work.

Author Response

The authors are thankful for the reviewer’s comments and suggestions. We addressed all the comments and suggestions below with reference on our revised manuscripts when possible (line numbers):

  1. Please consider spelling out the abbreviations (RFID and sUAS) in the article's title.

Response: The authors decided to update the title to “RFID and drones:…” We kept the RFID as it is more common but changed the sUAS as different countries have different names on drones, e.g., UAV, sUAS, multirotors, etc.

  1. "In the "Introduction" section, this section can be improved to provide further background and include all relevant references. There is a need that you will use "recent publications" on the topic to make your research attractive.

Response: We added four more publications in our Introduction to address this suggestion. The following references were added (Reference 15 ~ 18):

Bunker, R.; Elsherbeni, A. A Modular Integrated RFID System for Inventory Control Applications, Electronics 2017, 6, 6; http://doi.org/10.10.3390/electronics6010009

Herlin, A.; Brunberg, E.; Hultgren, J.; Högberg, N.; Rydberg, A.; Skarin, A. Animal Welfare Implications of Digital Tools for Monitoring and Management of Cattle and Sheep on Pasture. Animals 2021, 11, 829. https://doi.org/10.3390/ani11030829

Bridge ES, Wilhelm J, Pandit MM, Moreno A, Curry CM, Pearson TD, Proppe DS, Holwerda C, Eadie JM, Stair TF, Olson AC, Lyon BE, Branch CL, Pitera AM, Kozlovsky D, Sonnenberg BR, Pravosudov VV and Ruyle JE. An Arduino-Based

RFID Platform for Animal Research. Front. Ecol. Evol. 2019, 7:257. https://doi.org/10.3389/fevo.2019.00257  

Azmi, N.; Kamarudin, L.M.; Zakaria, A.; Ndzi, D.L.; Rahiman, M.H.F.; Zakaria, S.M.M.S.; Mohamed, L. RF-Based Moisture Content Determination in Rice Using

Machine Learning Techniques. Sensors 2021, 21, 1875. https://doi.org/10.3390/s21051875

And the following were added in Line 79~99

            Bunker and Elshebeni [15] developed a small portable RFID scanner comprised of a Raspberry Pi, and interrogator. They used a chip similar to what was used, but theirs in-cluded the development kit where their code will only interface directly to the USB port of the development kit. Their system was designed to be highly customizable and modular. Bridge et al. [16] developed similar system using an Arduino-based RFID platform where its application was for three animal applications (breeding behaviours of Wood Ducks, RFID-enabled bird feeder, and nest-box monitoring for breeding birds). Their system used an Arduino M0 and two RFID module, UB22270 from Atmel. The RFID modules used a simplified antenna, thin coil magnet wire, which can be easily created with an inductance between 1.25 ~ 1.3 mH. The antenna was placed on the entrance hole such that the move-ment of tagged ducks will be recorded. Although, the system was quite unique as it used a small RFID chip, its applications are very similar to RFID applications in manufacturing, where the interrogator is placed in one location. Recent work evaluating RFID and drones for monitoring and management of animals is reported [17], although the technology were used for different purposes. RFID was only used for position and tracking while the drone was used for counting animals using cameras. A study reported using RFID and wireless technology to predict the moisture content of rice [18]. Both the Received Signal Strength Indicator (RSSI) from the two wireless transceivers were used for predicting the moisture content in rice using Artificial Neural Network. They reported that both the RSSI of RFID and Zigbee transceivers can be used to predict the moisture content with an accuracy of more than 85%.

  1. In the “discussion” section, the author must extend the comparison between their approach and other ones that have been developed and used in the literature for the same or related purposes (I recommend increasing the number of Scientific articles cited, especially to compare the study context with similar studies). Also, in this section, the authors should also highlight the current limitations and usefulness of the proposed research, and briefly mention some precise directions that they intend to follow in their future research work.

Response: We have added four more publications in our introduction and presented a short discussion on the comparison of the two systems as compared to our system in Line 415 ~ 427:

      Two papers [15, 16] report on an RFID system similar to ours. Bunker and Elsherbeni [15] built an integrated RFID system using the same RFID module we used. Our studies used the M6e chip while they used the development kit board of M6e. Our objective was to minimize the size of our RFID box and focus in developing our own drivers for the M6e. Although, their system was portable enough, it was not intended to be used where the RFID reader will be moving during operation as compared to our application. Bridge et al. [16] developed their own Arduino-based system and used two RFID chips from Atmel. Their system was unique as it used an easily fabricated antenna. While their antennae was small, the power of their system was very limited as their application was confined to small space. Our effort summarizes critical foundational efforts that will be used in future field experiments at a large commercial nursery. We acknowledge a possible limitation of our current system in flight time, however, we envision improvements in battery technology will advance in the future which will address this limitation.

Round 2

Reviewer 2 Report

The application of methods of statistical analysis offer the opportunity to achieve a more accurate and more nuanced understanding of data. Unfortunately, the Authors might not yield reasonably accurate results, because of their choice to not apply any of them. This aspect continues to be the main criticism of the research.

Moreover, in introduction section the Authors inserted some interesting paper. But, generally the literature review does much more than just provide a list of research in similar topic and should summarise the most important result achieved of previously published work, identify gaps or unexplored areas. So they should rewrite L80-99.

L 143-192 - Are you sure that all the information provided in this paragraph are necessary?

i.e. The reader chip, m6e, is configured to use the highest power and work at a frequency of 900 Mhz.

i.e. The communication between the m6e and controller is through a serial port with 160 115200 baud rate.

i.e. powered throug a coin-cell battery (1.5V) located below the USB port

i.e. Time was used as part of the file 168 naming process for data whenever the system was powered.

i.e. All data are saved to a unique 169 file and stored in the microSD card.

i.e. The same data are also streamed through the transceiver port.

i.e. Data are typically stored in a comma separated values (CSV) file.

and so on

L325-326. In my modest opinion, the choice to combine the results of scan reading from five distance and four orientation without a statistical support, caused to lose of interesting consideration.

L340-341. They fall in the primitive error: I can not see in fig 11 what the Authors report. Where can I see that SR had the highest number of scanned tags?

Author Response

The authors are thankful for the reviewer’s comments and suggestions. We addressed all the comments and suggestions below with reference to our revised manuscripts when possible (line numbers). We also attached a pdf version of this response.

1. The application of methods of statistical analysis offer the opportunity to achieve a more accurate and more nuanced understanding of data. Unfortunately, the Authors might not yield reasonably accurate results, because of their choice to not apply any of them. This aspect continues to be the main criticism of the research.
Moreover, in introduction section the Authors inserted some interesting paper. But, generally the literature review does much more than just provide a list of research in similar topic and should summarise the most important result achieved of previously published work, identify gaps or unexplored areas. So they should rewrite L80-99.

Response: The authors have added Line 80~99 at the suggestion of one of the reviewers. Moreover, we also added a comparison and differences of these additional references under Section 4 (Discussion) at Line 415 ~ 427 as suggested by the same reviewer. We believe we addressed this issue in our last revision.

2. L 143-192 - Are you sure that all the information provided in this paragraph are necessary?
i.e. The reader chip, m6e, is configured to use the highest power and work at a frequency of 900 Mhz.
i.e. The communication between the m6e and controller is through a serial port with 160 115200 baud rate.
i.e. powered throug a coin-cell battery (1.5V) located below the USB port
i.e. Time was used as part of the file 168 naming process for data whenever the system was powered.
i.e. All data are saved to a unique 169 file and stored in the microSD card.
i.e. The same data are also streamed through the transceiver port.
i.e. Data are typically stored in a comma separated values (CSV) file.
and so on

Response: The authors' intention was to provide a thorough presentation of the Materials and Methods of our RFID Module. We believe providing such detail is important so other researchers can replicate this system.

3. L325-326. In my modest opinion, the choice to combine the results of scan reading from five distance and four orientation without a statistical support, caused to lose of interesting consideration.
Response: Results were combined based on the previous reviewer’s comments (below):
“In Mat & Met the Authors report that in view of the results of Experiment 1, four tags [9F (TDC), 6F (TDE), 5B (TDA) and 12 (TDI)] with the highest scan rate were chosen. Unfortunately, in the results of experiment 1 they did not report any information about this.”
Based on that input we updated the figures to show the top four tags on the preliminary laboratory test.

4. L340-341. They fall in the primitive error: I can not see in fig 11 what the Authors report. Where can I see that SR had the highest number of scanned tags?

Response: Similar to figures for Experiment 1, we updated figures 11 and 12 and show combined readings of the four tags (9F, 6F, 5B, and 12) to clearly highlight which distance yielded the highest and lowest scan total.

Reviewer 3 Report

The paper has been clearly improved.

As already mentioned in my previous review, please use straight borders for figures 3 and 7 (do not use curved borders).

In tables 6 and 7 the SD values are reported with too many decimal non meaningful digits: please keep just 2 decimal digits. 

Author Response

The authors are thankful for the reviewer’s comments and suggestions. We addressed all the comments and suggestions below with reference to our revised manuscripts when possible (line numbers):

1. As already mentioned in my previous review, please use straight borders for figures 3 and 7 (do not use curved borders).

Response: Thank you for the comments and suggestions. We updated both figures 3 and 7.

2. In tables 6 and 7 the SD values are reported with too many decimal non meaningful digits: please keep just 2 decimal digits.

Response: We updated Tables 6 and 7.
